# Genetic Aspects of Micronutrients Important for Inflammatory Bowel Disease

**DOI:** 10.3390/life12101623

**Published:** 2022-10-18

**Authors:** Sanja Dragasevic, Biljana Stankovic, Nikola Kotur, Aleksandra Sokic Milutinovic, Tamara Milovanovic, Milica Stojkovic Lalosevic, Maja Stojanovic, Sonja Pavlovic, Dragan Popovic

**Affiliations:** 1Clinic for Gastroenterology and Hepatology, University Clinical Center Serbia, Koste Todorovica Street 2, 11000 Belgrade, Serbia; 2Faculty of medicine, University of Belgrade, Pasterova Street 2, 11000 Belgrade, Serbia; 3Institute of Molecular Genetics and Genetic Engineering, University of Belgrade, Vojvode Stepe Street 444A, 11042 Belgrade, Serbia; 4Clinic for Allergy and Immunology, University Clinical Center of Serbia, Koste Todorovic Street 2, 11000 Belgrade, Serbia

**Keywords:** IBD, genetics, vitamin D, B9, B12, vitamin A, iron, zinc

## Abstract

Inflammatory bowel disease (IBD), Crohn’s disease (CD) and ulcerative colitis (UC) are complex diseases whose etiology is associated with genetic and environmental risk factors, among which are diet and gut microbiota. To date, IBD is an incurable disease and the main goal of its treatment is to reduce symptoms, prevent complications, and improve nutritional status and the quality of life. Patients with IBD usually suffer from nutritional deficiency with imbalances of specific micronutrient levels that contribute to the further deterioration of the disease. Therefore, along with medications usually used for IBD treatment, therapeutic strategies also include the supplementation of micronutrients such as vitamin D, folic acid, iron, and zinc. Micronutrient supplementation tailored according to individual needs could help patients to maintain overall health, avoid the triggering of symptoms, and support remission. The identification of individuals’ genotypes associated with the absorption, transport and metabolism of micronutrients can modify future clinical practice in IBD and enable individualized treatment. This review discusses the personalized approach with respect to genetics related to micronutrients commonly used in inflammatory bowel disease treatment.

## 1. Introduction

The pathogenesis of inflammatory bowel disease (IBD) highlights the role of mucosal immunology and changes in the gut microbiome triggered by genetic and environmental factors including diet regiments, as suggested by many nutritional studies [1,2,3,4]. Oxidative damage that occurs in CD and UC is a result of an altered balance between free radical production with antioxidant depletion and micronutrients, leading to antioxidant repletion [1,2,3,4]. The presence or absence of anti-inflammatory agents such as antioxidants obtained through dietary intake or supplementation can impact the course of IBD. The intestinal tissue damage and altered gut microbiota caused by oxidative stress are significantly impacted by the presence of tissue repair mediators. Modulating the intestinal microbiota remains an attractive therapeutic potential for IBD [1,2,3,4]. Changes in dietary habits were also found to be strongly associated with a determined increased risk of autoimmune disease in a pediatric population [5,6]. So far, dietary constituents have been considered precipitants or promoters of complex interactions in IBD pathology, while nutritional deficiency with imbalances of specific micronutrients has been associated with the course of the disease [1,2]. Nevertheless, the role of modifiable environmental and behavioral factors such as diet remains poorly understood.

The majority of IBD patients show an interest in the active management of their disease, especially through dietary modifications [7]. Specifically, long-term dieting has shown the most significant effect in shaping the intestinal microbiome [8]. Therapeutic strategies in IBD, along with medications, encompass nutritional interventions including not only the elimination of potential food triggers but also the improvement of the nutritional status of patients [1,2,9]. The supplementation of micronutrients and macronutrients is important in everyday clinical practice in reducing the primary or secondary symptoms of disease [2]. Nevertheless, overuse or treatment with doses far exceeding the recommended daily allowances can be harmful and lead to adverse effects on the course of IBD. Especially during the coronavirus (COVID-19) pandemic, the frequent use of over-the-counter supplements among IBD patients has contributed to inadequate and uncontrolled strategies in therapy management.

Nutritional guidelines based on genetic testing are still lacking, especially in IBD. One of the reasons for this is a lack of high-quality evidence on the effect of nutrition counseling based on genetics [10]. The high-throughput analysis of the human genome, based on next-generation sequencing (NGS) technology and genotyping microarrays, is becoming increasingly more available to researchers interested in nutritional genomics. NGS allows for the comprehensive analysis of the genome including rare variants and structural alterations. Genotyping arrays only cover a fixed number of single-nucleotide variants throughout the genome and can be used to analyze common genetic alterations. These technologies allow for the validation of already identified candidates and the search for novel genomic markers of micronutrient response in an unbiased manner.

### Complex Interface between Nutrition and Inflammatory Bowel Disease

The nutritional deficiencies in IBD patients could be associated with malabsorption, excess losses or decreased intake. Active inflammation in the gut is also related to increased metabolic needs, particularly active CD of the small bowel or in cases of intestinal resection [9,11].

The dysbiosis in IBD is characterized by a lower microbial diversity, the increased prevalence of pro-inflammatory microbes within the phylum *Proteobacteria*, and the reduced prevalence of commensal bacteria of the phylum *Firmicutes* [8,12].

Modifying the intestinal microbiome via nutritional factors can potentially modulate the IBD course of promoting intestinal inflammation by dysregulating the immune system, intestinal permeability, and the mucous layer [1]. Therefore, long-term dietary habits influence a final microbial composition in every individual. A “Westernized” diet rich in fat and sugar has been related to intestinal inflammation and gut dysbiosis characterized by an increase in *Proteobacteria* and a decrease in protective microbes [13,14]. Furthermore, a study that compared gut microbiomes of European and African children identified increased *Enterobacteriaceae* levels in children from Europe and an affluence of short-chain fatty acids (SCFA) associated with high plant polysaccharide consumption in Africans [15]. These observations indicated that SCFA-producing bacteria that are prominently detected in African children could prevent the establishment of pathogenic bacteria [15].

Various investigations have analyzed nutrient intake in IBD patients with diverse results. Mainly, these studies discovered inadequate intakes of fiber, vitamins C and D, vitamins B6 and B12, magnesium, calcium, β-carotene, phosphorus, and others among patients with Crohn’s disease and ulcerative colitis [7,16,17]. However, the methodological heterogeneity of the studies, the absence of healthy controls for nutritional intake, and limitations in assessment methodology using questionnaires led to inconsistent results in IBD [7].

According to current investigations, a third of patients with IBD have gut symptoms in the absence of the objective evidence of gastrointestinal inflammation characterized as irritable bowel syndrome [7,18]. Namely, non-inflammatory gut symptoms have been associated with dietary intake, while dietary triggers of gut symptoms have been reported in 60% of IBD patients [7,19].

Decreased appetite or aberrant absorption due to the epithelial defects of the gut cause nutrient deficiencies in IBD patients. Approximately over half of IBD patients, mostly diagnosed with CD, have a deficiency of essential vitamins and minerals [20]. Nutritional loss impacts intestinal epithelial tight junctions, microbial communities, and cellular changes associated with IBD occurrence [20]. Nevertheless, the question of whether nutritional deficit is a consequence or a risk factor for IBD remains.

In addition to the differences in diet and other environmental exposures, levels of essential micronutrients could be influenced by the variation in genes involved in absorption, transport or metabolic transformation determining individuals’ risk of deficiency or toxicity. Understanding the effects of those genetic variants may help identify subgroups who are more susceptible to micronutrient deficiencies or at risk of various health outcomes modified by those deficiencies. This will allow for the selection of subjects who are at an increased risk of nutritional deficiencies and could benefit trials of supplementation. Recommendations regarding genetically tailored micronutrient supplementation should also be in line with population genetics. Populations can carry different burdens for the risk of micronutrient deficiencies due to their specific genetic makeup; also, different genetic variants in diverse worldwide populations could be important for defining the risk of altered nutrient status, poor response to supplementation or disease occurrence.

This review describes micronutrients important for IBD pathogenesis and treatment, particularly vitamin D, folates, B12, vitamin A, iron, and zinc, as well as the genetic factors that contribute to their status. Further, we discuss current data related to the recognized genetic markers of altered micronutrient status and IBD. Our main goal was to present an overview of the available knowledge on selected micronutrients’ genetics that could be used as guidance for future research on IBD.

## 2. Role of Vitamins in Inflammatory Bowel Disease

### 2.1. Vitamin D

According to numerous investigations, the deficiency of vitamin D has been highlighted as a key factor in the pathogenesis of IBD (Table 1) [21,22]. Vitamin D is a liposoluble vitamin, and its hormonal form of 1,25-dihydroxy vitamin D3 [1,25(OH)2D3], also called calcitriol, is important for various pathways of the immune system mediated via nuclear vitamin D receptor (VDR) in immune cells such as T and B lymphocytes, monocytes and macrophages. Vitamin D has a role in immune cell differentiation, the modulation of the gut microbiota, gene transcription, and barrier integrity [22]. A reduction in the serum levels of vitamin D is associated with an increased risk for infection (Table 1) [21,22]. The role of vitamin D includes the support of intestinal epithelial junctions and the upregulation of junction proteins including claudins, ZO-1, and occludins. The disruption of the mucosal barrier was noted in an IBD investigation in polarized epithelial Caco-2bbe cells grown in a medium with or without vitamin D and challenged with adherent invasive *E. coli* strain (AIEC). The investigation showed that Caco-2bbe cells incubated with 1,25(OH)2D3 were protected against AIEC-induced disruption. Additionally, vitamin D-deficient mice with DSS-induced colitis showed significant increases in the quantities of *Bacteroidetes* and were more susceptible to AIEC colonization. According to previous studies, vitamin D contributes to the homeostasis of the intestinal barrier function and protection against adherent invasive *E. coli* [23]. Additionally, it has been suggested that patients with IBD are at an increased risk of *Clostridium difficile* infection. Vitamin D has a prophylactic role against infection, influencing the production of antimicrobial compounds such as cathelicidins and modulating the microbiome [24]. VDR regulates the biological action of 1,25(OH)2D3 and has a role in the genetic, immune, environmental and microbial aspects of IBD. Dionne at al. study indicated that 1,25(OH)2D3 in CD patients significantly decreases the proinflammatory activity of M1-type macrophage but does not provide a reduction in the anti-inflammatory actions of M2-type macrophages. The level of anti-inflammatory cytokine IL-10 was not affected in the investigation [25]. The deficiency of vitamin D is also correlated with disease activity in IBD patients, so administration targeting a concentration of 30 ng/mL could potentially reduce disease activity [22]. Even though reports have shown lower vitamin D levels in IBD patients compared with the healthy population, it is not clear yet if the vitamin D deficiency is a consequence of the disease itself or if it has a role in disease pathogenesis. A study that followed subjects in two time points before (up to 8 years) and one time point after IBD diagnosis showed that the vitamin D level was not altered in IBD patients prior to disease onset compared with matched controls, but it was reduced after the disease was established [26].

According to conducted studies, VDR regulates the function of T cells and Paneth cells while modulating the release of antimicrobial peptides in the gut interaction pathways. Furthermore, beneficial microbial metabolites, including butyrate, stimulate VDR signaling [22]. Ananthakrishnan et al. showed that one third of IBD patients included in their study had a vitamin D deficiency and that its decreased levels correlated with colon cancer incidence in patients with IBD [27]. For the most common confounders of vitamin D deficiency have been indicated low intensity of sunlight and dis-ease duration and activity [27]. Further studies determined that 30 ng/mL of serum circulated vitamin D form, 25(OH)D3, can inhibit the secretion of proin-flammatory cytokines (IL-6 and TNF-α) induced by lipopolysaccharide (LPS) of the bacterial wall [28,29].

### 2.2. Vitamin D-Related Genetics

It has been demonstrated in numerous candidate gene approach and genome-wide association studies (GWAS) that vitamin D status is partly determined by genetic factors. Several genes and genetic variants located in or near those genes, such as *DHCR7*, *GC*, *CYP2R1*, *CYP24A1* and *VDR*, have been recognized as significant modulators of vitamin D level and bioavailability [40]. Indicated genes encode proteins/enzymes involved in vitamin D transport and metabolism, namely, *DHCR7* encodes enzymes expressed in the skin that are involved in cholecalciferol synthesis, *GC* encodes a vitamin D-binding protein that has a role in vitamin D precursor transport, *CYP2R1* is a 25-hydroxylase involved in vitamin D precursor activation, and a *CYP24A1* encodes 24-hydroxylase that participates in the inactivation of vitamin D metabolites. Variants of *DHCR7* (rs12785878), *GC* (rs4588; rs7041), *CYP2R1* (rs10741657, rs1993116, and rs10766197) and *CYP24A1* (rs6013897) genes have been found to be associated with the serum level of 25-OHD [41,42], a form in which vitamin D is abundantly present in the circulation. The *VDR* gene encodes the vitamin D receptor, a transcription factor that regulates the expression of numerous genes after binding to the active form of vitamin D. Variants in this gene have been linked with the disease phenotype rather than with the level of the vitamin D.

It was demonstrated by Chip-seq that VDR-binding sites were significantly enriched near autoimmune-associated genes identified in GWAS, including the *PTPN2* gene linked to Crohn’s disease [43]. Research on different experimental models of the inflamed gut showed that intestinal epithelial VDR regulates the IBD-associated autophagy gene *ATG16L1* and lysozyme expression, as well as gut microbial assemblage—all important for maintaining the intestinal homeostasis [44]. Numerous studies have evaluated the association between IBD occurrence and genetic variants in the *VDR* gene since this gene maps to the region on chromosome 12 shown to be linked to IBD. The results regarding the association between *VDR* and IBD have been inconsistent, probably due to underpowered studies, different prevalences of vitamin D deficiency, and genetic diversity between different ethnic groups [26,45,46,47,48,49,50,51].

The FokI variant (rs2228570), which introduces an alternative translation start site, and three silent genetic variants of BsmI (rs1544410), ApaI (rs7975232), and TaqI (rs731236) in the *VDR* gene appear to be sporadically associated with IBD in diverse populations (Table 2). The association between the TaqI “t” (nucleotide C) allele and CD occurrence has been demonstrated in Caucasian populations [26,45,46,47]. Although the TaqI variant is a synonymous, “silent” variant located in the exon 9 of the *VDR* gene, it was shown that homozygous “tt” (genotype CC) carriers had significantly lower levels of the VDR protein in the PBMC of CD patients. The study also demonstrated that lower VDR levels were not associated with the changes in the mRNA expression nor with the production of the truncated protein [52]. In addition, CD carriers of the “tt” genotype exhibited a significantly higher risk (OR = 3.6) of having a B3-penetrating phenotype [52]. Regarding other variants, the results are not uniform; the FokI “f” (nucleotide T) variant was associated with CD in Iranian population [50] and with UC in Asian populations [47,51]; the presence of the BsmI “B” allele (nucleotide T) has been linked with an increased susceptibility to UC in Israeli Ashkenazi [48] and Han Chinese patients [49]; a meta-study indicated that carriers of the Apal “AA” genotype (TT genotype) had an increased risk for CD regardless of population stratification [47]. Overall, these results highlight the importance of examining population genetics in assessing disease burden or defining strategies for precision medicine/nutrition.

Compared with the *VDR* gene, only a few studies have assessed the association between IBD and other vitamin D-related genes, such as *DHCR7*, *CYP2R1* and *GC* [65,66]. They have previously been mostly examined in relation to vitamin D level and deficiency [41,42]. The role of those genes in IBD pathogenesis and treatment response should be more thoroughly analyzed in the future studies considering other factors that influence vitamin D level, such as vitamin D intake, nutrient supplement use, body mass index, physical activity, and lifestyle factors. The potential contribution of the *VDR* variants in response to the vitamin D treatment of IBD patients is promising. A recent meta-study conducted on a general population showed that *VDR* could play a role in the modulation of the response to vitamin D supplementation, showing that the FF genotype of the FokI variant and Tt + tt genotypes of the TaqI variant were associated with a better response to vitamin D supplementation [67].

### 2.3. Folate and Vitamin B12

Folate (vitamin B9) is a one-carbon moiety donor cofactor involved in nucleotide synthesis and methylation metabolic pathways. Due to an impaired methylation cycle, folate deficiency causes the accumulation of homocysteine, a metabolite associated with oxidative stress and inflammation (Table 1). Folate deficiency and elevated homocysteine levels are related to various pathological conditions such as anemia, low bone mineral density, thromboembolic events and birth defects [30].

In IBD, folate deficiency and elevated homocysteine levels are related to markers of intestinal inflammation [31,68,69], the reduced survival of regulatory T cells in the small bowel [70], and the active stage of the disease [31,32]. Folate deficiency may cause DNA hypomethylation, which is related to higher inflammation and colorectal cancer risk in IBD patients [71,72]. Results from a meta-analysis suggested that high folate levels can reduce the risk of IBD [73]. Due to these findings, it has been proposed that folate supplementation may be utilized to reduce complications of IBD [70,73].

Among IBD patients, around 20% have reduced folate levels and around 30% have increased homocysteine levels, which is much more common than in healthy people [31,74,75,76]. Bermejo et al. reported a higher prevalence of folate deficiency among CD patients (22%) compared with UC patients (4.3%), as well as an association with disease severity but not ileal resection (Table 1) [53]. Folate deficiency may be due to a reduced folate intake caused by the avoidance of folate-rich food, active inflammation that causes higher folate utilization, and the reduced absorption of folate due to intestinal damage, small bowel resection, or certain medications often prescribed to IBD patients (such as methotrexate and sulfasalazine) [77]. A recent meta-analysis showed that IBD patients consume an inadequate amount of cereals, legumes, fruit, vegetables, and dairy, which causes a lower intake of energy, calcium, fiber, and folate [78]. Vitamin B9 metabolites are absorbed in the proximal parts of the small bowel, so small bowel resection and severe intestinal inflammation related to IBD (which causes structural alterations of the bowel) reduce folate absorption [30]. To avoid folate deficiency, regular folate level monitoring and supplementation are recommended in IBD patients with a high risk of folate deficiency [79].

Vitamin B12 (cobalamin) and folic acid have significant roles in erythropoiesis and are often associated with anemia in patients with IBD [2,77]. Cobalamin and folate are crucial for nucleic acid synthesis and the process of erythropoiesis [2]. In the course of differentiation, erythroblasts need vitamin B12 and folic acid for proliferation, while their deficiency leads to macrocytosis, the apoptosis of erythroblasts, and anemia [2,77]. According to a previous investigation, the prevalence of vitamin B12 deficiency ranges between 6 and 38% [22,80]. Dietary vitamin B12 binds an intrinsic factor synthesized by the parietal cells in the duodenum for its further absorption in the terminal ileum. Hence, vitamin B12 deficiency is much more frequent in CD than in UC patients [33]. According to Battat et al., the crucial risk factor for vitamin B12 deficiency is an ileal resection of more than 30 cm [80]. Nevertheless, CD with ileal localization is not a risk factor for cobalamin deficiency [80]. A previous investigation determined that UC patients have a vitamin B12 deficiency similar to that of the general population [21]. However, UC patients with ileo-anal J-pouch could have vitamin B12 deficiency due to small bacterial overgrowth [22].

The recent guideline recommendations of the European Crohn’s and Colitis Organization (ECCO) endorsed checking folate and vitamin B12 levels at a minimum of once per year or when macrocytosis is present, especially in IBD patients not receiving thiopurines [81]. Even though folate deficiency and elevated homocysteine levels have been linked to IBD-associated colon cancer, preclinical studies with folic acid supplementation contradict previous published data [2,82].

### 2.4. Folate-Related Genetics

Besides the adequate intake and absorption of vitamin B9, optimal levels of folate bioactive forms and homocysteine are maintained by folate cycle enzymes and transporters. The activity of those proteins show marked inter-individual differences that depend on genetics. One of the key polymorphic enzymes of the folate cycle is methylenetetrahydrofolate reductase (MTHFR), which provides methyltetrahidrofolate, a bioactive form of vitamin B9 involved in re-methylation pathways. There are two common missense variants of the *MTHFR* gene that cause the reduced activity of the enzyme: c.677C>T (p.Ala222Val) and c.1298A>C (p.Glu429Ala), which have been extensively studied in IBD and other diseases.

Variant c.677T (rs1801133) causes a 70% reduction in enzyme activity [83]. Each copy of the low-activity T allele causes a greater reduction in folate and a higher level of homocysteine [84,85]. Initial reports found an elevated risk of IBD in carriers of the TT genotype, but the majority of subsequent studies, including high-quality studies, did not corroborate this finding [55,78]; however, further research is needed.

Variant c.1298A>C (rs1801131) causes a 40% reduction in enzyme activity [54], and it is less studied than the c.677C>T variant. The lower-activity c.677T and c.1298C variants very rarely lie on the same chromosome. However, in compound heterozygotes of the c.677C>T and c.1298A>C variants, higher homocysteine levels are expected [85]. Interestingly, a recent meta-analysis associated the presence of the lower-activity C allele of the c.1298A>C variant with a greater risk of UC and IBD [55] (Table 2). The same meta-analysis did not show compelling evidence that the c.677C>T variant is associated with IBD development.

Methotrexate is one of the immune-modulating drugs often prescribed to IBD patients. By inhibiting some of the key involved enzymes, methotrexate disrupts the folate cycle. This results in immune-modulating effects, including higher rates of the apoptosis of the immune cells and elevated levels of adenosine, a natural anti-inflammatory agent [86]. In addition to its therapeutic effects, methotrexate could also exert toxic effects, most notably on GIT, liver and bone marrow [87]. These side effects might be more common in patients with *MTHFR* variants [56,88], though more research is needed, especially studies involving IBD patients. Variants in other pharmacogenes, such as those involved in folate and methotrexate transport (most notably *SLCO1B1* and *SLC19A1*), might also play a role in response to methotrexate therapy in pediatric IBD patients and rheumatoid arthritis patients [89,90]. Methotrexate side effects can be mitigated by folate supplementation, so identifying patients at risk, such as those who carry unfavorable genetic variants, could be of great benefit.

### 2.5. Genetics of the Vitamin B12 Status

It is estimated that the heritability of B12 levels is considerable and reaches 59% [91]. Major findings on genetics related to B12 status were thoroughly described by Surendran et al. [92], identifying 59 significantly associated variants from 19 genes. The genetic component of B12 status seems complex; among B12-related genes are those included in vitamin B12 absorption (*FUT2*), transport (*TCN1* and *TCN2*), cellular uptake (*CUBN*), and the catalysis of enzymatic reactions in the one carbon cycle (*MTHFR* and *MTRR*).

One gene that has shown a strong and reproducible association with B12 levels is fucosyltransferase 2 (*FUT2*). *FUT2* encodes a secretor enzyme that adds fucose to oligosaccharides producing H-type antigens, precursors of blood group A and B antigens in the intestinal epithelium and in mucosal and salivary secretions [93]. It is known that 20% of the Caucasian population carries nonfunctional mutations resulting in “non-secretor phenotype”, meaning that homozygous individuals of those variants cannot express A, B, or H antigens in body fluids, which makes them prone to variety of gastrointestinal diseases, including IBD. Fucosylated glycans have a very important role in gut microbial composition, host–microbe interactions, and membrane stability [94]. The non-secretor nonsense variant *FUT2* rs601338, also known as G428A, was found to be strongly associated with B12 levels such that individuals with the non-secretor status variant had higher plasma vitamin B12 levels compared with carriers of the secretor status genotypes [95]. It has been suggested that reduced vitamin B12 absorption in carriers of the secretor genotype may be a consequence of the susceptibility of this phenotype to H. pylori infection and its related gastric-induced vitamin B12 malabsorption [95]. Interestingly, the *FUT2* gene is also a well-established susceptibility locus for CD, but in the case of CD, the non-secretor status increases the risk of the disease occurrence [57,96] (Table 2). This has been explained by a reduced microbial richness present in individuals with the non-secretor phenotype compared with the secretor phenotype [58,94]. The *FUT2* gene shows population-specific differences that should be taken into account in B12 deficiency risk assessment. For instance, the *FUT2* rs601338 variant is rare in Southeast and East Asian populations compared with European populations [92]. Another nonsynonymous *FUT2* variant, rs1047781 (A385T), is more frequent in East Asians; it is also related to reduced secretor phenotype levels and associated with vitamin B12 status [97].

Variants in the B12-binding protein genes transcobalamin I (*TCN1*) (rs526934, rs34528912, and rs34324219), transcobalamin II (*TCN2*) (rs1801198), and intestinal intrinsic factor receptor cubulin (*CUBN*) (rs1801222) have been also associated with circulating B12 levels [59,95,98,99]. A few studies analyzed the relationship between *TCN2* variants and predisposition to IBD; the association of the *TCN2* rs1801198 variant with CD and UC was not demonstrated [100,101,102]. One pathway analysis of GWAS CD data identified the rs9621049 variant in the *TCN2* gene as one of the potential contributors to ileal CD susceptibility in populations of European and Jewish ancestry [103].

### 2.6. Vitamin A (Retinol)

Vitamin A is a liposoluble vitamin with various forms including retinol, retinal, retinoic acid (RA) and carotenoids. The concentration of retinol in the serum is often used to identify vitamin A deficiency risk. Namely, a high proportion of IBD patients in the adult and pediatric population has been diagnosed with deficiency (Table 1) [34,35]. RA is an immunoactive form of vitamin A that increases T-cell proliferation [104]. Previously conducted studies underlined RA as a potential player in the improvement of CD8+ T cell immune response, as well as the increased secretion of IL-10 and TGF-β [105,106]. Nevertheless, it has been shown that the action of RA depends on its concentration in the body. Additionally, RA has effects on dendritic cell and gut mucosa CD103 and CXCR1 receptors that modulate intestinal macrophage homeostasis [107]. Interestingly, a case report described retinol deficiency and night blindness in a CD patient with repeated small bowel resections [36]; the regular parenteral administration of vitamin A restored normal eyesight in this CD patient [36]. Soares-Mota et al. determined that CD patients with vitamin A deficiency had significantly lower body mass index and body fat levels than those with normal levels [108]. So far, studies in human populations with the supplementation of vitamin A or RA have been disappointing [2]. However, the effectiveness of vitamin A supplementation could be potentially limited by the reduced expression of ALDH1a2 (Aldehyde Dehydrogenase 1 Family Member A2) and the increased activity of the RA-catabolizing enzyme CYP26A1 [109]. Nevertheless, the supplementation of vitamin A is indicated in cases with a confirmed deficiency to meet the recommended dietary allowance of 900 μg daily for men and 700 μg daily for adult women [2].

### 2.7. Genetic Variants Associated with Vitamin A Status

Genetic association studies have identified several genetic variants associated with the level of vitamin A, mainly present in the form of retinol and provitamin A carotenoids. A study that used the measurement of vitamin A level in liver samples indicated its association with the rs738409 (I148M) variant of the *PNPLA3* gene [110]. The *PNPLA3* gene encodes adiponutrin, a lipase involved in releasing retinol from lipid droplets in hepatic stellate cells. This variant was also associated with different chronic liver diseases including non-alcoholic fatty liver disease (NAFLD) and cirrhosis [111,112]. Additionally, the association of the *PNPLA3* I148M variant has been observed with fasting retinol serum levels in subjects with NAFLD and obesity [113]. NAFLD, a type of liver steatosis, is a common condition in IBD, but its development and progression are still poorly understood. Moreover, IBD patients experience higher mortality rates from NAFLD compared with the general population [114]. A study that examined two Italian cohorts showed that IBD carriers of the *PNPLA3* 148M allele have a higher risk of hepatic steatosis (odds ratio of 2.9) and higher biomarkers of liver damage [115] (Table 2). The distribution of the I148M *PNPLA3* variant considerably differs among populations (from 70% in Latin American populations to 20% in European and African populations) [60]. The monitoring of *PNPLA3* variant distribution among risk populations could contribute to reducing vitamin A deficiency and preventing different liver diseases in IBD patients and the general population.

Besides *PNPLA3*, other genes involved in the absorption, transport and metabolism of vitamin A have been linked to the blood level concentrations of this vitamin. A GWAS of 5000 Caucasian individuals found that two independent genetic variants, rs1667255 and rs10882272, were associated with circulating levels of retinol. The indicated variants are located near the genes encoding major carrier proteins of retinol—transthyretin (*TTR*) and retinol-binding protein 4 (*RBP4*) [116]. It has been shown that RBP4 and TTR levels are decreased during active inflammation despite a sufficient vitamin A status [117]. In one study that examined the relation between RBP4 and TTR serum levels and IBD in children, a negative correlation was shown between RBP4 and disease activity [118]. To the best of our knowledge, there are no data regarding the linkage of *RBP4* and *TTR* genetic variants and IBD. Another vitamin A-relevant gene was studied in the context of IBD—*CYP26B1*. The CYP26B1 enzyme is involved in the degradation of all-trans-retinoic acid, a vitamin A-active metabolite. One study analyzed the *CYP26B1* variant rs2241057, whose major allele T was previously shown to be linked to reduced vitamin A catabolism compared with the minor C allele. A higher frequency of rs2241057 TT carriers in CD compared with healthy controls was observed, meaning that the increased CD risk was linked to the genetically determined higher levels of active vitamin A [61]. The same study indicated an association of the TT genotype with non-stricturing, non-penetrating behavior in CD patients (Table 2). The authors speculated that this variant mediates changes in retinoic acid metabolism and thereby alters the induction of Th17 cells and the IL23/Th-17 pathway, which has a key role in IBD pathogenesis [119].

It is also worth mentioning the retinoic acid-inducible transcription factor ISX, which is an important regulator of provitamin A fluctuations in mammals. ISX is expressed in the intestine, and it downregulates the expression of two genes, *SCARB1* and *BCO1*, encoding for proteins that mediate the uptake of carotenoids and their conversion into retinoids [120]. Genetic variants in the *ISX, BCO1*, and *SCARB1* genes have been associated with beta carotene levels in humans [121,122]. Moreover, the *ISX* gene has been identified as a susceptibility gene for Crohn’s disease [123]. It has been suggested that ISX plays an important role in the homeostatic control of both vitamin A metabolism and immunity processes [124]. Therefore, it deserves further attention in the studies of nutrition genetics especially related to inflammatory gut diseases.

## 3. Minerals in Inflammatory Bowel Disease

### 3.1. Iron

Iron is an essential element found in myoglobin, hemoglobin, and transferrin, as well as other enzymes. It is crucial for blood production and reversible oxygen binding in the hemoglobin. Iron is partially ionized from the trivalent to divalent states in the stomach due to acidic conditions. Iron is absorbed in the small intestine and stored in the liver, kidney, blood serum, spleen and bone marrow. The bioavailability of heme iron is approximately up to 20%, and that of nonheme iron is up to 5%. The absorption of iron depends on food components, physical health, medications and dietary supplements [21].

The incidence of iron deficiency anemia (IDA) in IBD patients is high and often associated with fatigue, with a prevalence of up to 76% of patients (Table 1) [37]. The pathogenesis of IDA in IBD patients includes the inadequate intake of iron, malnutrition, chronic blood loss due to intestinal mucosa ulcerations, or inflammation secondary to aberrant iron absorption in the intestine. Iron represents an essential nutrient in nearly all bacterial species in the gut microbiota. Previously conducted studies on the role of oral iron in IBD are controversial. Although CD mouse models indicated the aggravation of disease activity after iron ingestion, parenteral use had no promoting effect [125]. Namely, microbial composition detected with luminal iron depletion showed a decrease in *Desulfovibrio* sp. Nevertheless, in the majority of human IBD studies, no difference was shown in disease activity between the use of oral iron and parenteral supplementation. So far, no studies have investigated the microbiome with respect to iron status in IBD, but African children with anemia after oral iron fortification showed an unfavorable ratio of fecal *Enterobacteriaceae* to *Bifidobacteria* and *Lactobacilli*, resulting in an increase in fecal calprotectin [126]. Furthermore, IL-6-driven increases in hepatic hepcidin, which binds to ferroportin on enterocytes and monocytes, result in its internalization, lysosomal degradation, and (finally) intracellular iron sequestration [21]. Guideline recommendations for IBD patients include 30 mg/day of elemental iron for IDA prophylaxis and 50–60 mg/day for treatment. Nevertheless, oral supplementation in IBD may be unproductive due to the normocytic anemia of chronic inflammation. Frequent adverse effects result in poor adherence to treatment, while high doses and an excess of non-absorbed iron in IBD may be toxic to the epithelium. The efficacy of oral iron is low in IBD patients with increased values of C-reactive protein, so oral supplementation is considered safer and more effective in patients with inactive or mild forms of IBD [20]. According to ECCO guidelines, intravenous iron should be considered the first line of treatment in patients with clinically active IBD, previous intolerance to oral iron, hemoglobin levels below 10 g/dL, or the documented need for erythropoiesis-stimulating agents [81].

### 3.2. Genetic Variants Associated with Iron Status

Although often considered to have an environmental etiology, iron status may be contributed by genetics as well. Multiple genetic variants have been shown to be associated with imbalanced iron-related biomarkers, resulting from iron overload or deficiency, particularly in genes involved in iron hemostasis pathways—*TMPRSS6*, *HFE*, *TF*, *HAMP*, *TFR2* and *SLC40A1* [127].

One of the most frequently reported genetic variants associated with lower iron levels is the missense variant rs855791 (A736V) in the *TMPRSS6* gene [128,129,130,131,132]. The association was consistent across various ethnic groups and replication cohorts, although it should be noted that the frequency of the *TMPRSS6* rs855791 variant considerably differs between non-African (~50%) and African (10%) populations [127]. The *TMPRSS6* encodes a membrane serine protease that can strongly suppress the expression of hepcidin, a peptide hormone that is the major negative regulator of iron levels in humans. Hepcidin is produced in the liver in response to iron excess and during acute or chronic inflammation. Since pathogenic bacteria require iron, one of the major defense strategies against infection involves the stimulation of hepcidin expression by pro-inflammatory cytokines, but in the case of prolonged inflammation, this can cause iron-restricted erythropoiesis and anemia [133]. It was shown that *TMPRSS6* A736V is a functional variant and that 736A inhibits hepcidin more efficiently than 736V [134]. On the other hand, variants associated with iron excess have been found in the homeostatic iron regulator gene *HFE*, transferrin gene *TF*, and transferrin receptor 2 gene *TFR2* [132,135,136]. The most commonly reported iron status-related *HFE* variant is rs1800562 (C282Y), which is widely associated with a severe form of hereditary haemochromatosis in European populations [137]. It has been estimated that the *HFE* C282Y variant and three variants in *TF* (rs3811647, rs1799852, and rs2280673) can explain ~40% of the genetic variation in serum transferrin levels [137].

The role of genetic variants related to iron levels and iron deficiency anemia in IBD has been poorly investigated. It is speculated that iron status is associated with the development of certain inflammatory responses and inflammatory diseases. One Mendelian randomization study measured the causal associations of iron status with gout, rheumatoid arthritis, and inflammatory bowel disease using *HFE* and *TMPRSS6* genetic variants as instrumental variables for exposure [138]. A study found that genetically predicted high levels of ferritin and transferrin saturation and low levels of transferrin were positively associated with gout and inversely associated with rheumatoid arthritis, while association with IBD was not supported (Table 2). Two studies analyzed *TMPRSS6* and *HFE* variants in adult and pediatric celiac disease in which, as in IBD, iron deficiency anemia is a very common condition [62,63]. Interestingly, the studies demonstrated an association of *HFE* C282Y with iron deficiency anemia, in contrast with the expected role of this variant in iron overload. Additionally, the authors suggested the potential contribution of the *TMPRSS6* rs855791 variant in predicting responses to oral iron supplementation in celiac disease patients affected by iron deficiency anemia [63] (Table 2). More studies are needed to reveal the role of iron-related genes in the pathogenesis of IBD. Moreover, understanding genetic factors that contribute to iron levels might direct the future implementation of iron intervention strategies in IBD treatment.

### 3.3. Zinc

Zinc is an essential mineral that plays many crucial roles in aspects of cellular metabolism, including supporting the catalytic activity of approximately 100 enzymes, the modulation of immune function, protein synthesis, wound healing, DNA synthesis, cell division, and the improvement of intestinal barrier function. Zinc levels and the assessment of its status significantly fluctuate with intake due the fact that zinc lacks storage mechanisms. So far, it has been estimated that 15% of IBD patients have zinc deficiency [38]. Zinc deficiency has been associated with altered functions of the digestive system in IBD. Mammals do not possess adequate zinc storage, so daily intake is essential. Zinc-enrichment processes by Lactobacilli and Bifidobacteria strains can improve the bioavailability of this essential metal in vivo. The oral supplementation of zinc-enriched probiotics has demonstrated the inhibition of *Escherichia coli* infection in mice. Additionally, previously conducted investigations determined the protective effects of probiotics, such as *Lactobacillus reuteri* and *Bifidobacterium bifidum*, against UC in humans and animals [139]. Siva et al. demonstrated an association of zinc deficiency in CD and UC patients with poor clinical outcomes, an increased risk of subsequent hospitalizations, surgeries, and disease-related complications (Table 1) [39]. Additionally, the results of the study indicated that the IBD outcomes improved with the normalization of zinc values, and the authors suggested the close monitoring and supplementation of zinc in IBD patients with 11 mg/day for males and 8 mg/day of elemental zinc for females [39]. Nevertheless, in some studies, higher doses have been used in IBD patients from 40 mg/day for 10 days to 110 mg three times a day for 8 weeks in CD patients in remission [140]. The high-dose and long-term supplementation of zinc should be implemented with caution due to side effects/toxicity and interference with iron and copper absorption that lead to their deficiencies.

### 3.4. Genetic Variants Associated with Zinc Status

Data regarding genetics that modify zinc status are limited. Candidate gene studies have mostly been focused on genes encoding two families of zinc transporters, ZnT (*SLC30A*) and ZIP (*SLC39A*), which are engaged in reducing and increasing zinc levels, respectively, in the cytosolic compartment; also, genes encoding zinc-binding proteins, metallothioneins (MTs), have been considered important [140,141]. Several genetic variants in zinc transporters and MTs have been associated with type 2 diabetes/impaired glucose regulation (*SLC30A8* rs13266634 and rs11558471), chronic diseases associated with aging (*MT2A* rs10636 and rs1610216), and cognitive performance (*SLC30A3* rs73924411) [130]. Moreover, it has been shown that *SLC30A8* variants are good candidates for modifying zinc requirements [142,143,144].

In addition to the relationship of variants in zinc transporter genes with various health outcomes, they have been also found to be associated with blood/serum zinc levels (*SLC30A3* rs11126936) [145,146]. A GWAS on 2603 Australian and 2874 British study participants identified three loci associated with zinc blood levels, and none of these involved genes for the zinc transporters but did include the genes for carbonic anhydrases (*CA1, CA2, CA3,* and *CA13*), phosphopantothenoylcysteine decarboxylase (*PPCDC*)*,* and Zn-finger proteins (*KLF8, ZXDA,* and *ZXDB*) [147]. Regarding the genetics of zinc status and IBD occurrence, there are not much currently available data. A large GWAS on 10,523 IBD cases and 5726 controls found a missense variant rs13107325 (A391T) in the zinc transporter gene *SLC39A8* associated with CD [148]. The variant was also associated with disease location and behavior in CD, as well as with shifts in the composition of gut microbiota in both CD and healthy subjects (Table 2). The authors speculated that the rs13107325 variant could disturb the transmembrane domain of SLC39A8, which might alter zinc metabolism and affect innate and adaptive immunity, as well as the gut microbiota [64]. A subsequent study replicated this association [148], while a study that used the murine model for *Slc39a8* A391T demonstrated that these mice exhibited an increased sensitivity to epithelial injury and pathological inflammation in the colon [149]. In a trans-ethnic meta-analysis of genome-wide data from a cohort of 86,682 European individuals and 9846 individuals of non-European descent, *SLC30A7* was highlighted as being among candidate genes associated with UC occurrence [150]. The identified association of IBD and genetic variants in zinc transporters, as well as the important role of zinc in the regulation of gastrointestinal homeostasis and adequate immune response, indicate that more studies on zinc biology and IBD pathogenesis are needed, especially randomized trials of genetically guided zinc supplementation in IBD patients.

## 4. Conclusions

Genetic variants important for vitamin D, folates, B12, vitamin A, iron and zinc status have been more or less identified in GWAS and candidate gene studies; however, their contribution to the treatment and susceptibility of IBD needs to be more thoroughly investigated. The overlap between the risk associated with the altered status of micronutrients and IBD occurrence have been observed for variants in the *VDR, FUT2, PNPLA3* and *SLC39A8* genes. This could be important for dealing with IBD patients within populations with higher frequencies of those genetic variants in order to predict and prevent the possible deficiencies closely related to IBD pathogenesis, therapeutic failures, or the occurrence of IBD following comorbidities such as NAFLD. Overall, the individualization of micronutrients’ supplementation in IBD patients with respect to genetics is a promising but still not feasible practice. In the following years, high-throughput technologies [151] and their affordable price will hopefully enable the identification of reliable markers of micronutrient supplementation valuable for IBD prediction, efficient treatment, or even prevention.

## Figures and Tables

**Table 1 life-12-01623-t001:** Frequent deficiencies of micronutrients in IBD.

Micronutrient	Signs of Deficiencies	Risk Factor for Deficiencies in IBD	Laboratory Values of Deficiencies	References
Vitamin D	Hypocalcemia, osteomalacia, osteoporosis.	Ileal localization of CD; resection of ileum > 20 cm; significant gastric resection; SIBO; diet.	25(OH)D < 20 ng/mL.	[22,26]
Folate (B9)	Megaloblastic, macrocytic anemia; diarrhea; nervous instability; dementia.	Restrictive diet in IBD; GI resections; therapy of Sulfasalazine and methotrexate; SIBO.	Low serum levels of folate; elevated MCV and homocysteine.	[30,31,32]
Vitamin B12	Megaloblastic, macrocytic anemia; tiredness. mouth ulcers; muscle weakness; disturbed vision; psychological problems.	More frequent in CD than in UC patients; ileal resection > 30 cm; gastric resection; SIBO.	Low serum levels of vitamin B12.	[22,33]
Vitamin A	Dry eye and skin; night blindness; infertility; delayed growth; throat and chest infections; poor wound healing; acne.	CD patient with repeated small bowel resections.	Low serum levels of vitamin A	[34,35,36]
Iron	Microcytic, hypochromic anemia; tachycardia; fatigue; sleepiness; headache; anorexia; nausea; pallor.	GI bleeding, more frequent in UC, achlorhydria, SIBO.	Low serum levels of iron, serum ferritin < 100 ng/mL, transferrin saturation < 20%, elevated transferrin receptor levels.	[21,37]
Zinc	Altered growth, hypogonadism, impaired night vision, anorexia, diarrhea alterations in taste and smell, alopecia, impaired wound healing.	CD fistulizing disease, PPI/H2 blockers, protein deficiency, malabsorptive disorders, diarrhea, restrictive diet.	Low plasma/serum zinc	[38,39]

SIBO—small intestinal bacterial overgrowth (SIBO); GI—gastrointestinal.

**Table 2 life-12-01623-t002:** Genetic variants that are associated with micronutrient levels and are potentially relevant to IBD pathogenesis, comorbidities or treatment.

Micronutrient	Genetic Variant Common Name	Variant ID	Nucleotide Change	Effect Allele	Functionality of the Effect Allele	Association of Effect Allele with IBD	Studied Population(s)	References
Vitamin D	*VDR* FokI	rs2228570	T>G	T	Associated with higher transcriptional activity of VDR	Associated with CD and UC occurrence	Iranian, Asian	[47,50,51]
	*VDR* BsmI	rs1544410	G>T	T	Unclear	Increased susceptibility to UC	Israeli Ashkenazi, Han Chinese	[48,49]
	*VDR* ApaI	rs7975232	G>T	T	Unclear	Increased risk to CD	European, Asian	[47]
	*VDR* TaqI	rs731236	T>C	C	Associated with lower transcriptional activity of VDR	Associated with CD occurrence; increased risk for penetrating phenotype	Caucasian	[26,45,46,49,52]
Folate (B9)	*MTHFR* 677	rs1801133	C>T	T	Reduction in MTHFR enzyme activity that provides a bioactive form of vitamin B9	Possible elevated risk for IBD in carriers of TT genotype	Meta-analysis	[53,54]
	*MTHFR* 1298	rs1801131	A>C	C	Reduction in MTHFR enzyme activity that provides a bioactive form of vitamin B9	Greater risk of UC and IBD; more likely to experience side effects of methotrexate compared with patients with the wild-type genotype	Meta-analysis	[55,56]
Vitamin B12	*FUT2* W143X	rs601338	G>A	A	Associated with the ABO non-secretory phenotype and higher levels of B12	Susceptibility locus for CD	Caucasian	[57,58]
	*FUT2* I129F	rs1047781	A>T	T	Associated with reduced ABO secretor phenotype and higher B12 levels	Association with CD	Asians	[59]
Vitamin A	*PNPLA3* I148M	rs738409	C>G	G	Associated with vitamin A levels and chronic liver diseases (NAFLD)	IBD carriers of the 148M allele have a higher risk of hepatic steatosis and higher biomarkers of liver damage	Two Italian cohorts	[60]
	*CYP26B1* L264S	rs2241057	T>C	T	Linked to reduced vitamin A catabolism compared with the minor C allele	Higher frequency of TT carriers in CD compared with healthy controls	Swedish	[61]
Iron	*TMPRSS* 6736V	rs855791	T>C	C	Associated with lower iron levels	Not found to be associated with IBD in one Mendelian randomization study; potential contribution in predicting response to oral iron supplementation in celiac disease patients affected by iron deficiency anemia	European	[62,63]
	*HFE* C282Y	rs1800562	G>A	A	Associated with increased iron levels and severe form of hereditary haemochromatosis in European populations	Not found to be associated with IBD in one Mendelian randomization study; associated with iron deficiency anemia in celiac disease patients	European	
Zinc	*SLC39A8* A391T	rs13107325	G>A	A	Located in zinc transporter gene; might alter zinc metabolism	Associated with CD, disease location/behavior, shifts in the composition of gut microbiota in both CD and healthy subjects	Netherlands	[64]

## Data Availability

Not applicable.

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
