# Peer review of "Genetic Aspects of Micronutrients Important for Inflammatory Bowel Disease"

_life, 2022, doi:10.3390/life12101623_

Round 1
Reviewer 1 Report
Thank you for the possibility to review the manuscript entitled “Genetic aspects of micronutrients important for inflammatory bowel disease”. The subject matter discussed in the article is current and very important. The available knowledge on selected micronutrients genetics which could be used as a guidance for the future research in IBD has been well analysed. The article needs to be only slightly corrected before it can be accepted for publications.
Comments to the Author:
1. Why didn’t you describe selenium which is also very important micronutrient for IBD pathogenesis and treatment?
2. Have the studies have been conducted in paediatric population? Please mention it.
3. Line 509: I don’t understand this sentence. Please clarify it. What does it mean “genetically modified zinc supplementation in IBD is needed”?
4. Please clearly state whether we are able to individualize the use of supplements in patients in our daily practise with respect to genetics related to micronutrients at the moment?
5. Conclusion is too long. Please shorten it
Author Response
Our response:
We thank the Reviewer for taking the time to read our manuscript. We have answered all the questions raised by the Reviewer, point by point. All changes in the manuscript are highlighted in red.
Comments to the Author:
1. Why didn’t you describe selenium which is also a very important micronutrient for IBD pathogenesis and treatment?
Our response:
Although many conducted studies found that selenium deficiency is frequent in IBD, therapy and management of UC and CD is challenging in everyday clinical practice due to numerous adverse effects of different treatments. Selenium has anti-inflammatory properties, contributes to the antioxidant system, mucosal healing and improves gut protective microbiota. Nevertheless, selenium supplementation can cause a number of adverse effects including diarrhea, nausea, vomiting, unusual tiredness and weakness etc. similar to the symptoms of IBD relapse. Namely, selenium supplementation in IBD has shown promising results in preliminary experimental and clinical studies, but results remain too limited, that is why selenium was not included in our comprehensive review. Randomized clinical trials are needed to measure the short-term and long-term effects of selenium on active and quiescent phase of IBD especially in patients with documented selenium deficiency.
2. Have the studies have been conducted in paediatric population? Please mention it.
Our response:
Data regarding pediatric population, if available, have been also included in the manuscript. Precisely, pediatric cohorts have been described in the following lines or have been highlighted in the revised manuscript if this was not explicitly mentioned:
[Introduction, lines 45-47] “Changes in dietary habits were also strongly associated with the determined increased risk of autoimmune disease in the pediatric population [5, 6].”
[Folate-related genetics, lines 321-324] “Variants in other pharmacogenes, such as those involved in folate and methotrexate transport, most notably SLCO1B1 and SLC19A1, might also play a role in response to methotrexate therapy in pediatric IBD patients as well as rheumatoid arthritis patients [75, 76].”
[Vitamin A (retinol), line 368-369] “Namely, a high proportion of IBD patients in the adult and pediatric population has been diagnosed with deficiency (Table 1) [93, 94].”
[Genetic variants associated with vitamin A status, lines 413-415] “In one study that examined the relation between RBP4 and TTR serum levels and IBD in children, a negative correlation was shown between RBP4 and disease activity [110].
[Iron, lines 507-509] “Two studies analyzed TMPRSS6 and HFE variants in adult and pediatric celiac disease in which, as in IBD, iron deficiency anemia is a very common condition [133, 134].”
3. Line 509: I don’t understand this sentence. Please clarify it. What does it mean “genetically modified zinc supplementation in IBD is needed”?
Our response:
We modified the indicated sentence to be clearer for the readers, as follows:
[Lines 571-575] “Identified association of IBD and genetic variants in zinc transporters, as well as the important role of zinc in the regulation of gastrointestinal homeostasis and adequate immune response indicate that more studies on zinc biology and IBD pathogenesis are needed, especially randomized trials of the genetically guided zinc supplementation in IBD patients.”
4. Please clearly state whether we are able to individualize the use of supplements in patients in our daily practice with respect to genetics related to micronutrients at the moment?
Our response:
We clearly stated our opinion regarding individualization of the supplement’s usage in IBD patients in the Conclusion section.
[Lines 585-587] “Overall, individualization of micronutrients’ supplementation in IBD patients with respect to genetics is promising but still not feasible practice.”
5. Conclusion is too long. Please shorten it.
Our response:
As Reviewer suggested, we have modified the Conclusions section to be more concise.
[Lines 577-590] “Genetic variants important for the vitamin D, folates, B12, vitamin A, iron and zinc status have been more or less identified in GWAS and candidate gene studies, however, their contribution to the treatment and susceptibility of IBD needs to be investigated more thoroughly. Overlap between the risk associated with altered status of the micronutrients and IBD occurrence have been observed for the variants in genes VDR, FUT2, PNPLA3 and SLC39A8. This could be important for dealing with IBD patients within populations with higher frequencies of those genetic variants in order to predict and prevent the possible deficiencies tightly related to the IBD pathogenesis, therapeutic failures or occurrence of IBD following comorbidities such as NAFLD. Overall, individualization of micronutrients’ supplementation in IBD patients with respect to genetics is promising but still not feasible practice. In the following years high-throughput technologies [136] and their affordable price will hopefully enable identification of reliable markers of micronutrient supplementation valuable for the IBD prediction, efficient treatment or even prevention. Also, more data from randomized control trials of the genetically guided micronutrient supplementation is needed to support their individualized usage in IBD treatment in the future.”

Reviewer 2 Report
This review describes micronutrients important for IBD pathogenesis and treatment, particularly vitamin D, folates, B12, vitamin A, iron and zinc, as well as the genetic factors that contribute to their status. It was to present an overview of the available knowledge on selected micronutrients’ genetics which could be used as a guidance for the future research in IBD.
The whole manuscript is clear in thinking, reasonable in explanation, and the cited literature is well-founded, which provides a good genetic research direction for the study of the pathological mechanism of IBD. Clinical research guidance has far-reaching significance.
Recommend it!
Author Response
Our response:
We thank the Reviewer for taking the time to carefully read and comment on our manuscript.

Reviewer 3 Report
The current study has a relevant topic and is within the aim and scope of the journal. It is of interest to the readers of this journal. The manuscript is clearly written in professional, unambiguous language throughout. The manuscript summarizes the recent studies on micronutrients, including vitamin D, folates, vitamin B12, vitamin A, iron, and zinc, that are related to IBD pathogenesis and treatment, genetic factors that contribute to their status and markers of micronutrient status and IBD that have been recognized. I have minor comments below.
In table 2, please clarify that the population is “population studies” as these are summarized from other publications.
Please briefly discuss the high-throughput technology that can be used to enable the identification of reliable markers of micronutrient supplementation valuable for IBD prediction, efficient treatment, or even prevention (line 527).
Author Response
Our response:
We thank the Reviewer for taking the time to read our manuscript. We have answered all the questions raised by the Reviewer, point by point. All changes in the manuscript are highlighted in red.
- In table 2, please clarify that the population is “population studies” as these are summarized from other publications.
Our response:
In Table 2, we modified “Population” into “Studied population(s)”.
- Please briefly discuss the high-throughput technology that can be used to enable the identification of reliable markers of micronutrient supplementation valuable for IBD prediction, efficient treatment, or even prevention (line 527).
Our response:
In the introduction section, a paragraph on high-throughput technology application in the field nutritional genomics is added.
[Lines 64-73] “Nutritional guidelines based on genetic testing are still lacking, especially in IBD. One of the reasons for that is lack of high-quality evidence on the effect of nutrition counseling based on genetics [9]. High-throughput analysis of the human genome, based on next generation sequencing technology and genotyping microarrays is increasingly more available to researchers interested in nutritional genomics. NGS allows comprehensive analysis of the genome including rare variants and structural alterations of the genome. Genotyping arrays cover only a fixed number of single nucleotide variants throughout the genome, and can be used to analyze common genetic alterations. These technologies allow validation of already identified candidates and search for novel genomic markers of micronutrient response in an unbiased manner.”

Reviewer 4 Report
The title of this article is “Genetic aspects of micronutrients important for inflammatory bowel disease”. This is an interesting topic. However, there are still some areas of the article that need to be revised.
1. The article "Complex interface between nutrition and inflammatory bowel disease". The authors mention the relationship between intestinal flora and inflammatory bowel disease, which is an important point, and the authors need to broaden the introduction, such as the improvement of inflammatory bowel disease through the intake of different nutrients to regulate intestinal flora.
2. The authors mentioned the association of Folate on inflammatory bowel disease, for this part, the authors need to introduce more experimental cases to support the point.
3. The article "Genetic variants associated with vitamin A status" section. This section is interesting and the authors need to analyze it in more depth and introduce more references to illustrate the link between vitamin A and genetic variants.
4. The authors analyze the alterations that occur in minerals in inflammatory bowel disease, and for this part, the authors need to go deeper and give more of their own perspective as well as an outlook for the future.
5. Authors are requested to carefully check the format of the references used in the article to ensure that the references are in the required format.
Author Response
Our response:
We thank the Reviewer for taking the time to read our manuscript. We have addressed all the comments and suggestions given by the Reviewer, point by point. All changes in the manuscript are highlighted in red.
- The article "Complex interface between nutrition and inflammatory bowel disease". The authors mention the relationship between intestinal flora and inflammatory bowel disease, which is an important point, and the authors need to broaden the introduction, such as the improvement of inflammatory bowel disease through the intake of different nutrients to regulate intestinal flora.
Our response:
We introduced new paragraph in the Introduction section and sections related to the specific micronutrients:
[Introduction, Lines 39-45]: “Oxidative damage that occurs in CD and UC is a result of altered balance between free radical production with antioxidant depletion and micronutrients leading to antioxidant repletion. Presence or absence of anti-inflammatory agents such as antioxidants obtained through dietary intake or supplementation can impact the course of IBD. Intestinal tissue damage and altered gut microbiota caused by oxidative stress is significantly impacted by the presence of tissue repair mediators. Modulating the intestinal microbiota remains an attractive therapeutic potential for IBD.”
[Vitamin D, Lines 139-152]:” Role of vitamin D includes support of intestinal epithelial junctions, upregulation of junction proteins including claudins, ZO-1, and occludins. Disruption of the mucosal barrier has been noted in IBD investigation in polarized epithelial Caco-2bbe cells grown in medium with or without vitamin D and challenged with adherent invasive E. coli strain (AIEC). The investigation showed that Caco-2bbe cells incubated with 1, 25(OH) 2D3 were protected against AIEC-induced disruption. Also, vitamin D deficient mice with DSS induced colitis had significant increases in the quantities of Bacteroidetes and were more susceptible to AIEC colonization. According to previous studies, vitamin D contributes to homeostasis of the intestinal barrier function and protection against adherent invasive E. coli [PMID: 25590952]. Also, it has been suggested that patients with IBD are at increased risk of Clostridium difficile infection. Vitamin D has a prophylactic role against infection, influencing production of antimicrobial compounds like cathelicidins and modulating microbiome [PMID: 25206277]”
[Iron, Lines 450-460]: “Iron represents an essential nutrient in nearly all bacterial species in the gut microbiota. Conducted studies on the role of oral iron in IBD are controversial. Although CD mouse models indicated aggravation of disease activity after iron ingestion, parenteral use had no promoting effect [PMID: 21076126]. Namely, microbial composition detected with luminal iron depletion showed decrease in Desulfovibrio sp. Nevertheless, in majority of human IBD studies no difference was shown in disease activity between the use of oral iron and parenteral supplementation. So far, there are no studies investigating microbiome with respect to iron status in IBD, but African children with anemia after oral iron fortification showed unfavorable ratio of fecal Enterobacteriaceae to Bifidobacteria and Lactobacilli, resulting with an increase in fecal calprotectin [PMID: 20962160].”
[Zinc, Lines 523-530]: “Zinc deficiency has been associated with altered functions of the digestive system in IBD. Mammals don't possess adequate zinc storage, therefore daily intake is essential. Zinc-enrichment processes by Lactobacilli and Bifidobacteria strains can improve the bioavailability of this essential metal in vivo. Oral supplementation of zinc-enriched probiotics demonstrated inhibition of Escherichia coli infection in mice. Also, conducted investigations determined the protective effects of probiotics, such as Lactobacillus reuteri and Bifidobacterium bifidum against UC in humans and animals [PMID: 31498347].”
- The authors mentioned the association of Folate on inflammatory bowel disease, for this part, the authors need to introduce more experimental cases to support the point.
Our response:
We added more information (in red) on folate intake, levels and association with inflammatory bowel disease:
[Lines 253-254; Lines 255-257] “In IBD, folate deficiency and elevated homocysteine level are related to markers of intestinal inflammation [46, 47, 48], reduced survival of regulatory T cells in the small bowel [49], and the active stage of the disease [48, 50]. Folate deficiency may cause DNA hypomethylation, which is related to higher inflammation and colorectal cancer risk in IBD patients [51, 52]. Results from a meta-analysis suggested that high folate levels can reduce risk of IBD (PMID: 31014995). Due to these findings, it is proposed that folate supplementation may be utilized to reduce complications of IBD [49, 53], however, no such recommendations have been adopted because high quality evidence from randomized controlled trials are still lacking.
[Lines 265-272] “Among IBD patients, around 20% have reduced folate level, and 30% increased homocysteine level, which is much more common than in healthy people [48, 54-56]. Bermejo et al. registered higher prevalence of folate deficiency among CD patients (22%) compared 231 to UC patients (4.3%) and association with disease severity, but not ileal resection (Table 1) [57]. A recent meta-analysis (PMID: 34323292) showed that IBD patients consume an inadequate amount of cereals, legumes, fruit, vegetables and dairy, which causes lower intake of energy, calcium, fiber, and folate. Vitamin B9 metabolites are absorbed in the proximal parts of the small bowel, so, small bowel resection and severe intestinal inflammation related to IBD, which causes structural alterations of the bowel, reduce folate absorption [PMID: 34836291]. To avoid folate deficiency, regular folate level monitoring and supplementation is recommended in IBD patients with high risk of folate deficiency [PMID: 30137275].”
- The article "Genetic variants associated with vitamin A status" section. This section is interesting and the authors need to analyze it in more depth and introduce more references to illustrate the link between vitamin A and genetic variants.
Our response:
We introduced new paragraph in the "Genetic variants associated with vitamin A status" section (lines 429-438):
“Worth mentioning is also the retinoic acid-inducible transcription factor ISX which is an important regulator of provitamin A fluctuations in mammals. ISX is expressed in the intestine and it downregulates expression of two genes, SCARB1 and BCO1, encoding for proteins that mediate the uptake of carotenoids and their conversion into retinoids [PMID: 25701869]. Genetic variants in the ISX, BCO1 and SCARB1 genes have been associated with the beta carotene levels in humans [PMID: 26063065; PMID: 22113863). Moreover, the ISX gene has been identified as a susceptibility gene for Crohn's disease [PMID: 23071489]. It has been suggested that ISX plays an important role in the homeostatic control of both vitamin A metabolism and immunity processes [PMID: 29073082]. Therefore, it deserves further attention in treating gut diseases with dietary interventions.”
- The authors analyze the alterations that occur in minerals in inflammatory bowel disease, and for this part, the authors need to go deeper and give more of their own perspective as well as an outlook for the future.
Our response:
We have covered the topic related to minerals and IBD as much as we could to make it informative but also concise. Our outlook for the future is described in the Conclusion section where we indicated that we need more data from genetic studies and randomized control trials to support the individualized usage of micronutrients in IBD treatment.
- Authors are requested to carefully check the format of the references used in the article to ensure that the references are in the required format.
Our response:
As Reviewer suggested, we have corrected, added and checked the format of the references used in the article to ensure that the references are in the required format.
